# Optimizing the Preparation of Silk Fibroin Nanoparticles and Their Loading with Polyphenols: Towards a More Efficient Anti-Inflammatory Effect on Macrophages

**DOI:** 10.3390/pharmaceutics15010263

**Published:** 2023-01-12

**Authors:** Antonio José Ruiz-Alcaraz, María Ángeles Núñez-Sánchez, María Alejandra Asensio Ruiz, María Antonia Martínez-Sánchez, Alba Oliva-Bolarín, Teresa Martínez Martínez, José Julián Pérez Cuadrado, Bruno Ramos-Molina, Antonio Abel Lozano-Pérez

**Affiliations:** 1Department of Biochemistry, Molecular Biology B and Immunology, School of Medicine, University of Murcia, Regional Campus of International Excellence “Campus Mare Nostrum”, 30100 Murcia, Spain; 2Obesity and Metabolism Laboratory, Biomedical Research Institute of Murcia (IMIB), 30120 Murcia, Spain; 3Radiopharmacy Research Group, Biomedical Research Institute of Murcia (IMIB), 30120 Murcia, Spain; 4Department of Biotechnology, Genomic and Plant Breeding, Murcian Institute for Agricultural and Environmental Research (IMIDA), 30150 Murcia, Spain

**Keywords:** nanoparticles, silk fibroin, polyphenols, curcumin, quercetin, resveratrol, inflammation, macrophages, IL-6, TNF-α

## Abstract

Silk fibroin nanoparticles (SFN) have become a promising tool in drug delivery systems due to their physicochemical characteristics. SFN have shown their outstanding properties as an active vehicle for polyphenols, enhancing their antioxidant and anti-inflammatory effects on macrophages; therefore, it becomes necessary to have an easy, reproducible and scalable production method. In order to improve the production of nanoparticles, we performed direct precipitation of non-dialyzed silk fibroin solutions and evaluated the reproducibility of the method using dynamic light scattering. We also studied the loading efficiency of three different natural polyphenols using propylene glycol as a solvent. The loaded nanoparticles were fully characterized and used to treat human macrophage cells to assess the anti-inflammatory activity of these nanoparticles. The measured hydrodynamic characteristics of the SFN and the overall yield of the process showed that the new preparation method is highly reproducible and repeatable. Thus, we not only present a new scalable method to prepare silk nanoparticles but also how to improve the loading of natural polyphenolic compounds to the SFN, as well as the important anti-inflammatory effects of these loaded nanoparticles in a cell model of human macrophage cells.

## 1. Introduction

Silk fibroin nanoparticles (SFN) have become a promising tool in drug delivery systems. They represent a highly customizable material with great potential in biomedical applications due to their mechanical and physicochemical properties [1]. Currently, there are multiple methods available to prepare SFN from two opposite approaches [2,3,4,5]. On one hand, the “top-down” methods are based on the size reduction in the fibroin mainly through mechanical milling [6]. However, these types of methods produce a large heterogeneity and low reproducibility in particle size [7]. On the other hand, the “bottom-up” approaches are based on the self-assembly of the proteins from the regenerated silk fibroin (SF) aqueous solution [8,9], the water-in-oil emulsion solvent evaporation [10], lipid templating [11], laminar jet breakup [12], spray drying [13], two-phase microfluidic flow-focusing devices [14], self-assembly from regenerated silk fibroin ionic liquid solution [15] or via solution-enhanced dispersion by supercritical CO_2_ [16]. The interest in microfluidic methods has recently increased [17,18,19] due to the homogeneity and reproducibility of the particle sizes compared to traditional methods based on precipitation in hydro-alcoholic solvents [9,20]. However, scaling up microfluidic-based methodology is still complex. Compared to the conventional bulk method, the currently purposed SFN production method is in a continuous manner, with significantly reduced size and batch-to-batch variation, which are critical factors for potential clinical translation [17]. Additionally, the method based on the precipitation of dialyzed aqueous solutions of fibroin is time-consuming and requires large volumes of water to remove the necessary salts for the dissolution of natural silk fibers. As the production of SFN for biomedical applications is expected to increase in the coming years [21], having a fast and scalable production method is mandatory.

The most common process used for the preparation of nanoparticles is the precipitation of the aqueous fibroin solution in an anti-solvent mixable with water (i.e., methanol, ethanol or acetone), which causes the change in conformation of the fibroin from random coil to β-sheet and the formation of nanoparticles [9]. However, this process starts from the SF solution, which is obtained after the long process of SF dissolution in either 9.3 M lithium bromide (LiBr) or the Ajisawa’s solvent system (CaCl_2_/EtOH/H_2_O 1:8:2) and their subsequent dialysis [8,22]. However, in a previous study carried out by dissolving fibroin in ionic liquids (such as 1-butyl-3-methylimidazolium chloride ([BMIM^+^][Cl^−^]), we showed the successful precipitation of the fibroin from an aqueous solution without the need of a previous dialysis step [15], which translates in reducing the total time of the process.

Similarly, there are several studies reporting the suitability of SFN as active vehicles for the transport and release of biomolecules targeting cancer cells [23,24] or to treat inflammatory conditions such as inflammatory bowel disease [25,26,27,28] and periodontitis [29]. However, the achievement of high drug loading content (DLC) becomes challenging due to the inherent nature of the biomolecules as well as the fibroin one. SFN are known to be stable in most solvents used for the dissolution of such biomolecules. Nevertheless, when the distribution coefficient between the nanoparticle surface and the solvent is low, it is required to increase the concentration of biomolecules in order to obtain suitable levels of DLC. Thus, using a low-toxicity solvent with a high dissolution capacity for the biomolecules and miscible in water will allow the biomolecules to become insoluble on the surface of the nanoparticles in a more efficient manner; therefore, to increase the DLC leading to a better therapeutic potential.

One interesting suitable solvent for molecules with low polarity is propylene glycol (PG), or 1,2-propanediol, which is a colorless and odorless liquid and has been established to be biologically safe. Although this small hydroxyl-substituted hydrocarbon (C_3_H_8_O_2_) is used mainly as a chemical intermediate in the production of unsaturated polyester resins and plasticizers, it is also commonly used as an excipient in a variety of drugs formulations, and its use in food and cosmetics as an additive has been authorized [30]. Indeed, it has many other applications, i.e., it has been used as an antifreeze or de-icing liquid, and added to latex paints and coatings to improve freeze–thaw capability. PG is included in the list of food additives generally regarded as safe (GRAS) by the US Food and Drug Agency and is considered to raise negligible concern for adverse effects on development and reproduction by the US National Toxicology Program (NTP) Center for the Evaluation of Risks to Human Reproduction (CERHR) [31]. It is also accepted for use as a food additive (E-1520) in Europe [32]. PG is commonly used as a solvent in many pharmaceuticals, including oral, injectable and topical formulations, which contain pharmaceutical drugs that are insoluble in water. Thus, natural polyphenols (PPh), such as curcumin (CUR), quercetin (Q) or resveratrol (RES), which present very low solubility in water, can be efficiently dissolved in PG and mixed with the SFN suspension in order to facilitate their adsorption onto the SFN.

The aim of this study was to describe a new, optimized, scalable and reproducible method to prepare silk nanoparticles, to demonstrate their capacity to load natural polyphenolic compounds as well as to test the anti-inflammatory effects of these loaded nanoparticles in a cell model of human macrophage-like cells. In order to improve the nanoparticles obtaining process, we first evaluated the effect of direct precipitation of SF solutions in 9.3 M LiBr and assessed the reproducibility of the method by dynamic light scattering (DLS) characterization. We also studied the load efficiency of three different PPh: CUR, Q and RES, using PG as a solvent. Loaded nanoparticles were sterilized by gamma-irradiation and used to treat human macrophage cells obtained from the differentiation of HL-60 cells, in order to evaluate the anti-inflammatory activity of these nano-formulated natural compounds.

## 2. Materials and Methods

### 2.1. Chemicals

All the chemicals and solvents used were purchased from Merck (Madrid, Spain). Ultrapure water (18.2 MΩ·cm^−1^) purified with an ELGA Purelab Flex 2 (High Wycombe, UK) was used.

### 2.2. Preparation of the Silk Fibroin Solution

SF was purified from white silk cocoons from *Bombyx mori* silkworms reared in the sericulture facilities of the Murcian Institute for Agricultural and Environmental Research (IMIDA, Murcia, Spain), which were fed with fresh natural *Morus alba* L. leaves. Cocoons were opened by scissors and the chrysalides were removed prior to being degummed in a boiling aqueous solution of 0.05 M Na_2_CO_3_ for 120 min, in order to efficiently remove the sericin while offering suitable characteristics for nanoparticle production [33]. The SF fibers were further rinsed with ultrapure water and dried at room temperature overnight. Then, SF was dissolved at 10% (*w*/*v*) in 9.3 M LiBr for 3 h at 65 °C, as previously described [34]. The amber-like SF solution was then filtered through two layers of Miracloth (Millipore, Merck, MA, USA) in order to remove remaining fibers or dust particles and stored at 4 °C until use.

### 2.3. Preparation of the Silk Fibroin Nanoparticles

SFN were prepared using a new nanoprecipitation method evolved from our previously described method [35]. Briefly, for nanoparticle preparation, the un-dialyzed SF LiBr-aqueous solution was diluted with ultrapure water to 1% SF (*w*/*v*) and slowly dripped into vigorously stirred methanol. After the complete addition of the SF, the nanoparticle suspension was stirred for another 2 h to complete the transition to β-sheet, and the obtained nanoparticles were recovered by centrifugation at 8000× *g* for 15 min at 8 °C (Eppendorf Centrifuge 5810R, Eppendorf AG, Hamburg, Germany). The pelleted particles were repeatedly washed with water in order to remove the methanol and the LiBr (until conductivity was lower than 10 μS·cm^−1^). Each washing cycle started with the suspension of the pelleted SFN aided by high-power ultrasounds for 1 min at 10% of amplitude in a Branson Digital Sonifier SFX 550 equipped with a 1/8″ tapered microtip (Branson Ultrasonics Corp, Danbury, CT, USA) and centrifugation at 16,000× *g* for 45 min at 8 °C. Then, the SFN were dispersed in ultrapure water, and the concentration of nanoparticles was measured by weighting dried replicates of known volumes of the SFN suspension (n = 3). Finally, SFN prepared for reproducibility determination were adjusted to 10 mg·mL^−1^ with ultrapure water, aliquoted in vials freeze-dried at −55 °C for 72 h in a Christ Alpha 1–2 LDPlus (Martin Christ Gefriertrocknungsanlagen GmbH, Osterode am Harz, Germany). The yield of the new method of production was calculated by applying the following equation:
(1)$$\mathrm{Yield}\;\left(\%\right)=\frac{\mathrm{weight}\;\mathrm{of}\;\mathrm{recovered}\;\mathrm{SFN}}{\mathrm{weight}\;\mathrm{of}\;\mathrm{dissolved}\;\mathrm{SF}}\times 100\;\left(\%\right)$$

SFN prepared for the biological assays, either naked or loaded with PPh, were sterilized by γ-irradiation, as previously described [21], and stored at 4 °C until use.

### 2.4. Polyphenols Loading on Silk Fibroin Nanoparticles

SFN were loaded with PPh, such as CUR, Q or RES, by an incubation method adapted from our previously described protocol [26]. Briefly, for the optimization of the nanoparticle loading, the aqueous suspensions of nanoparticles at 10 mg·mL^−1^ were diluted to the required concentrations (0–8 mg·mL^−1^) with ultrapure water and mixed with a solution of the PPh in PG at 5 mg·mL^−1^ to give a final volume of 1 mL with ratios PPh/SFN of 2:1, 1:1, 1:2, 1:4, 1:8, 1:16 and 1:32 (*w*/*w*) and a final composition water/PG 80:20 (*v*/*v*). Controls containing only SFN and free PPh were included and incubated in the same conditions. After the incubation period was completed, the nanoparticle suspensions were stirred for a further 24 h to complete the adsorption and the resulting nanoparticles were recovered by centrifugation at 16,000× *g* for 15 min at room temperature (Eppendorf Centrifuge 5810R, Eppendorf AG, Hamburg, Germany). Supernatants were separated from pelleted loaded nanoparticles, and both were stored at 4 °C until their characterization and analysis.

### 2.5. Nanoparticle Characterization

The characterization of the nanoparticles was performed using common techniques, such as field emission scanning electron microscopy (FESEM), DLS, attenuated total reflectance-Fourier transformed infrared spectroscopy (ATR-FTIR) and ultraviolet–visible spectroscopy following the procedure described previously [35].

The hydrodynamic diameter of nanoparticles (Z_ave_) and Z-potential (ζ) were determined by DLS using a Zetasizer Nano ZSP instrument (Malvern Instruments). All measurements were carried out using a disposable folded capillary Zeta cell containing 0.7 mL of the nanoparticle suspension in NaCl 1 mM at 20 °C according to the manufacturer’s instructions. Zeta potential was calculated from the electrophoretic mobility of nanoparticles using the Smoluchowski approximation. All measurements were performed in quintuplicate in NaCl 1 mM solution at 25 °C.

The morphology of the nanoparticles was observed by using a FESEM APREO S (Thermo Fisher Scientific Inc., Waltham, MA, USA). An aliquot (10 µL) of an aqueous suspension of nanoparticles (10 µg·mL^−1^) was dropped onto a clean glass wafer before drying overnight and finally sputtered with platinum for 5 min resulting in a 5.13 nm coating thickness (Leica, EM ACE600, Leica Microsystems Inc., Concord, ON, Canada). The morphology was studied by collecting images at a magnification of 50,000× using a T3 detector in the immersion mode (current 0.10 nA, accelerating voltage of 5.00 kV, WD = 4.5–5.0 mm).

Attenuated total reflectance-Fourier transformed infrared spectroscopy (ATR-FTIR) analysis of the nanoparticles was performed to detect structural changes in SF after PPh loading. Briefly, infrared spectra of ~2 mg of the freeze-dried samples were acquired on a Nicolet iS5 spectrometer, equipped with an iD5 ATR accessory (Thermo Scientific, USA) controlled by OMNIC Software Ver. 9.7.39. Measurements were made in absorbance mode, 64 scans at a resolution of 4 cm^−1^, in the spectral range of 4000–550 cm^−1^ and using N–B strong apodization and Mertz phase correction. Automatic Baseline Correct (corrected from 4000.1221 to 550.0952 cm^−1^; polynomial order: 2 and number of iterations: 20) and advanced ATR-correction were applied to the spectra (crystal: diamond) (refractive index = 2.40; sample refractive index = 1.60; angle of incidence = 45.0°; number of bounces = 1.0).

Spectroscopy studies were performed in a Synergy MX spectrometer (BioTek Instruments Inc.; Winooski, VT, USA) by using Corning 96-well polystyrene medium bind strip well microplates (Corning, NY, USA). Absorbance spectra were recorded in water in the range 300–550 nm (step 5 nm).

### 2.6. Determination of the Polyphenolic Loading Content and Encapsulation Efficiency

The polyphenols loading content (PLC) and encapsulation efficiency (EE) of the PPh in the SFNs were determined by spectrophotometric analysis following the previously described methods [24,25,26].

The PLC (%) and EE (%) of each PPh in the SFN were determined by an indirect method [36]. The quantity of PPh loaded onto nanoparticles was obtained as the difference between the initial quantity in solution and the remaining after incubation with SFN. The PLC and EE were obtained using Equations (2) and (3), respectively:
(2)$$\mathrm{Polyphenolic}\;\mathrm{Loading}\;\mathrm{Content}\;\left(\mathrm{PLC}\right)=\frac{\mathrm{weight}\;\mathrm{of}\;\mathrm{PPh}\;\mathrm{in}\;\mathrm{nanoparticles}}{\mathrm{total}\;\mathrm{weight}\;\mathrm{of}\;\mathrm{loaded}\;\mathrm{nanoparticles}}\times 100\;\left(\%\right)$$
(3)$$\mathrm{Encapsulation}\;\mathrm{efficiency}\;\left(\mathrm{EE}\right)=\frac{\mathrm{weight}\;\mathrm{of}\;\mathrm{PPh}\;\mathrm{in}\;\mathrm{nanoparticles}}{\mathrm{weight}\;\mathrm{of}\;\mathrm{PPh}\;\mathrm{in}\;\mathrm{the}\;\mathrm{loading}\;\mathrm{solution}}\times 100\;\left(\%\right)$$
where the weight of PPh in nanoparticles was calculated as the difference between the initial amount of PPh added and the remaining in solution after the incubation and centrifugation measured by spectrophotometry. Calibration curves (See Appendix A), in the range 0–1000 μg·mL^−1^, were prepared in water/PG (9:1 ratio), and absorbance was measured at the wavelength of the absorption peaks of each PPh, as previously described [24,25,26].

### 2.7. Stock Preparation for In Vitro Assays

Free PPh (CUR, Q and RES) were dissolved in dimethyl sulfoxide (DMSO) at a concentration of 10 mg·mL^−1^. Working solutions at a concentration of 1000 μg·mL^−1^ for each of the tested nanoparticles were prepared from the SFN-PPh suspensions at 10 mg·mL^−1^ (sterilized by γ-irradiation at 2.5 kGy as previously described [21]) by dilution with sterile phosphate-buffered saline (PBS), pH = 7.4 (Biowest, Nuaillé, France). Serial dilutions of the stocks were made at the appropriate concentration in complete cell-culture medium (CCM) prior to being used in the in vitro assays. Compounds were finally added to the cell wells to obtain the final assay concentrations. Empty nanoparticles were used as controls.

### 2.8. Cell Line and Culture Conditions

Free and SFN-PPh were tested using macrophage cells obtained from differentiation of the human acute myeloid leukemia cell line HL-60 (ATCC^®^ CCL-240™, American Type Culture Collection, Rockville, MD, USA) as the selected in vitro model. Cells were maintained in CCM consisting of RPMI-1640 (Biowest, Nuaillé, France), supplemented with 10% fetal bovine serum (GIBCO Invitrogen, Paisley, UK) and 1% penicillin/streptomycin (GIBCO), at 37 °C with 5% CO_2_ before differentiation.

All in vitro assays were performed after passage number 5 and before passage number 20 with cells growing at an exponential ratio. Cells were plated at 1 × 10^5^ cells/well (96-well plates) or 1 × 10^6^ cells/well (12-well plates) in CCM containing 10 ng·mL^−1^ phorbol myristate acetate (PMA) (Sigma Chemical Co., St. Louis, MO, USA) for a period of 24 h to allow differentiation. Cells were then rested in CCM without PMA for another 24 h prior to the addition of the different treatments.

### 2.9. In Vitro Viability Assays

Cytotoxicity was determined by MTT assay (based on the reduction in MTT (3-(4,5-dimethylthiazol-2-yl)-2,5-diphenyltetrazolium bromide) to formazan), as previously described, with small modifications [15]. In brief, after differentiation of human HL-60 to macrophage cells with PMA, cells were treated with either 100 μg·mL^−1^ or 500 μg·mL^−1^ SFN-PPh sterilized by γ-irradiation (2.5 kGy). Considering the PLC of nanoparticles, these concentrations of nanoparticles were equivalent to a final content of either 16 μg·mL^−1^ or 80 μg·mL^−1^ of each of the tested compounds (equivalent to 43.5 and 217 μM of CUR; 53.0 and 264 μM of Q; 70.0 and 350.5 μM RES). Next, after 1 h of incubation, LPS (*Escherichia coli 0111*. B4; Sigma Chemical Co.) was added at a final concentration of 0.1 μg·mL^−1^ to the media as low-degree pro-inflammatory stimuli, and cells were incubated for another 24 h. MTT (Alfa Aesar, Thermo Fisher, Karlsruhe, Germany) was added at a final concentration of 483 μM (0.2 mg·mL^−1^), and cells were further incubated for 1 h at 37 °C and 5% CO_2_. Afterwards, an acidified isopropanol solution containing 0.04 M hydrochloric acid and 0.1% NP-40 detergent was added to each well to lysate cells to dissolve the purple formazan. The optical density (OD) of the colored solution was measured at 570 nm using a multimode microplate reader (SPECTROstar Nano, BMG LABTECH, Ortenberg, Germany). Non-treated cells as well as cells treated with either unloaded SFN, non-irradiated SFN-PPh or free PPh were also included as control conditions.

### 2.10. In Vitro Anti-Inflammatory Assays

Supernatants from the in vitro assays of macrophage cells were measured to determine the levels of pro-inflammatory cytokines TNF-α and IL-6 using enzyme-linked immunosorbent assay (ELISA) kits following the manufacturer’s instructions (eBiosciences, San Diego, CA, USA), as previously described by Ruiz-Alcaraz et al. [37]. The absorbance in each well was measured with a microplate reader (Spectrostar Nano; BMG Labtech, Ortenberg, Germany) at 450 nm and corrected at 570 nm. Concentrations of cytokines secreted by the differentiated macrophage HL-60 cells were obtained by using the corresponding standard curve, and the modulation of cytokines production was measured and calculated as the level of release inhibition compared to control conditions. Cytokine concentration was normalized for each singular experiment by using the negative control (untreated cells) of the corresponding assay as a referential point (considered as 1 for control conditions). Finally, the effects upon cytokines’ levels were displayed as the mean ± SEM of normalized results.

### 2.11. Determination of Polyphenol-Loaded Silk Fibroin Nanoparticles Internalization by Fluorescent Microscopy

To confirm the internalization of SFN-PPh loaded by macrophages, the distribution of SFN inside the cells was determined by fluorescent microscopy analysis (DMi8, Leica Microsystems, Wetzlar, Germany). SFN-CUR was selected to perform these assays as CUR naturally emits green fluorescence and can be detected under the 488 nm laser [37]. HL-60 cells were seeded in 12-well plates at a density of 10^6^ cells/well in CCM with 10 ng·mL^−1^ PMA to allow differentiation. Treatment with SFN-CUR at a concentration of 16 μg·mL^−1^ was carried out as previously described. After the treatment, cells were washed 3 times with PBS, and fluorescence was detected using the 488 nm excitation laser. Pictures were taken in at least 5 fields per well at 10× magnification. Pictures were analyzed using the Leica Application Suite X (*LAS X*) software (Leica Microsystems, Wetzlar, Germany).

### 2.12. Statistical Analysis

Data are reported as normalized mean ± SEM, using the negative control (untreated cells) as the reference level. Statistical differences were analyzed by Student’s *t*-test or ANOVA test. Values of *p* < 0.05 were considered to show statistical significance. Calculations were performed using GraphPad Prism 8.0.1 and SPSS 21.0 software.

## 3. Results and Discussion

### 3.1. Preparation of the Silk Fibroin Nanoparticles

When the dissolved SF—in either 9.3 M LiBr [34] or Ajisawa’s solvent system [38]—was poured into an excess of a polar organic solvent, such as methanol, a milky white suspension of protein nanoparticles immediately appeared. While SF is maintained in the solution, peptidic chains are present in a random coil conformation due to multiple interactions of the side chains of the fibroin with the salt ions and the water molecules. It has been previously discussed how a fast replacement of their solvation sphere by an anti-solvent, such as methanol, induces the rapid folding of the peptide chains towards a more stable β-sheet assembly [39]. Although the classical methods of SF regeneration in the form of nanoparticles achieved by pouring an aqueous SF solution into an excess of a polar organic solvent are time-consuming due to the long dialysis step [1,9], it has generally been considered an essential stage of the process because it is routinely applied in other preparations of silk biomaterials [22]. However, taking into account our previous research with SF regenerated from an ionic liquid solution [15], where the dialysis step was not implemented, we proposed here the same methodology for SFN preparation. Thus, the SF chains should be able to reconstitute the hydrogen bonds network and thus change from a random coil structure to the highly ordered β-sheet conformation in the solid particles. In order to achieve this desolvation more efficiently, a previous stage of dilution of the SF solution was introduced, according to the previously published results of Lammel et al. (2010), who showed that lower concentrations of SF produce smaller particles [39]. This shortcut in the preparation of the nanoparticles, avoiding the dialysis step, is shown in Figure 1.

As shown in Figure 2a, the milky white suspension of SFN is formed immediately after the SF solution is dropped into the methanol under gentle stirring (See Appendix A). The rapid dissolution was favored not only by the stirring, but also by the lower concentration of SF dissolved in LiBr as well as the low viscosity of the diluted LiBr solution (similar to those of the dialyzed SF aqueous solution). With this new method, the undesirable effect of the dialyzed SF solution instability upon gelation during storage [40] was also avoided because the SF solution in 9.3 M LiBr can be stored at 4 °C without gelation for over 3 months. Then, SFN in suspension were easily recovered by centrifugation, and either salt (LiBr) or methanol were efficiently removed by washing with ultrapure water (Figure 1). The washing steps were repeated until the conductivity of the supernatants was lower than the required value according to the subsequent use of the nanoparticles. We established it at 10 μS·cm^−1^.

In order to assess the reproducibility of the new SFN preparation method, three independent assays were performed. Three operators on different days prepared three SF solutions in 9.3 M LiBr following the procedure described in Figure 1. Once the SF was dissolved, the solutions were divided into six replicates/batch for their further precipitation in methanol. All the suspensions appeared after a few drops, and the turbidity increased with the amount of SF solution added. In all cases, the nanoparticles were efficiently pelleted after centrifugation and easily separated from the solvents. Then, the SFN were easily desalted by repeated washing cycles of approximately 1 h each cycle, which supposes a reduction in the processing time. Finally, the measured hydrodynamic characteristics of the SFN and the overall yield of the process showed that the new method is highly reproducible and repeatable (Figure 2b,c). Statistic parameters are listed in Appendix A.

Furthermore, the overall yield of the process after recovering the SFN as a dried powder after freeze-drying was 85.2 ± 3.7% (n = 18). A significant part of the unrecovered mass of SF was lost at the centrifugation stage in the washing steps because the smallest particles were unable to be pelleted in the conditions we applied (16,000× *g*). However, this yield might be improved by using higher *g*-forces or other separation methods.

### 3.2. Polyphenols Loading on Silk Fibroin Nanoparticles

Previously published results have reported some difficulties in obtaining high payloads of natural PPh, such as CUR, Q or RES [24,25,26], when common adsorption techniques are used. In order to improve both PLC and EE, the water/PG combination was chosen due to the complete miscibility of both solvents and the characteristics of PG as a solvent for PPh. The complete miscibility of PG and water favored the rapid interchange of the PG solvation sphere of the PPh to a less favorable 90:10 water/PG mixture. The desolvation of the PPh and the presence of the SFN aided the insolubilization of the PPh on the hydrophobic surface of the fibroins, thus improving the loading content. All the loaded nanoparticle suspensions were stable and showed intense fluorescence under UV light (See Appendix A).

In order to understand the effect of the ratio PPh/SFN on the loading and the hydrodynamic properties of the nanoparticles, different PPh/SFN mixtures were tested ranging from 0–200% in weight, as described in Section 2.4. The obtained results showed that the loading behavior of the selected PPh was similar in all cases but with slight differences in the DLS characteristics. Among them, SFN-CUR showed the highest absolute values of Z_ave_, PdI and z, with significant changes with respect to the unloaded SFN (Tukey, *p* < 0.05). Q loading also produced an increase in Z_ave_ and PdI, but, in contrast to CUR loading, it showed a reduction in the ζ.

As shown in Figure 3a, the PLC values increased with the increment in the ratio Q/SFN. While the increment was roughly linear at lower ratios, it reached a plateau above the 1:4 ratio, which showed a PLC of 19.31 ± 2.35% and an EE of 16.17 ± 1.56%. Higher values did not produce further improvement in the loading parameters (i.e., higher %PLC and %EE). The hydrodynamic characteristics of the different loaded nanoparticles in this experiment were consistent with the PLC behavior (Figure 3). The diameters (Z-average) of SFN-Q displayed a rapid increase with an increment in the ratio Q/SFN up to the 1:8 ratio (12.5% in weight), which presented a diameter of ~170 nm. Higher ratios of Q/SFN were not translated into a proportional increase in size. On the contrary, a plateau was reached showing that the maximum loading of SFN-Q was reached. The polydispersity (PdI) and ζ values also varied with the ratio Q/SFN. PdI showed a parallel behavior increasing with the increment of the PLC loading.

The measured ζ shifted to lower absolute values with increasing PPh loading. These results agreed with those previously described [26] and showed that the loading procedure using a mixture of water/PG as the loading solvent system was appropriate because the three hydrodynamic characteristics of the loaded SFN were not affected.

Considering these results, a loading ratio Q/SFN of 1:4 was selected as the most appropriate for the optimal PPh loading in SFN on a higher scale (25 mg of PPh/100 mg of SFN). Thus, SFN-PPh (-CUR, -Q or -RES) were prepared for further cytotoxicity and anti-inflammatory assays in differentiated macrophage HL-60 cells.

Once the nanoparticles were loaded with CUR, Q and RES, before being used for the biological tests, their suspensions at 10 mg·mL^−1^ in ultrapure water were irradiated at 2.5 kGy for sterilization as previously described [21] without appreciable changes in the color or aggregation state. Only a statistically significant reduction in the Z_ave_ of SFN-CUR (Tukey, *p* < 0.05) was appreciated after the sterilization. Slight shifts on the other hydrodynamic parameters, such as a reduction in the PdI and an increase in the absolute ζ value, would indicate that the irradiation produced a slight disaggregation of the larger particles, which is translated to a lower Z_ave_ and PdI (narrower size distribution and more negative *ζ* value).

DLS characterization measurements of the nanoparticles used in the biological test confirmed the increase in the size of the loaded nanoparticles when compared with the unloaded SFN measured during the optimization assay. The hydrodynamic parameters of the aqueous suspensions of the SFN-PPh are depicted in Figure 4. All samples showed nanometric size, low PdI and a moderately negative ζ with a narrow distribution.

Thus, while SFN-CUR showed a noticeable increase in Z_ave_ and PdI in both cases with respect to the unloaded SFN (*p* < 0.0001, Dunnett), the effect on the ζ value was less intense in the irradiated samples (*p* < 0.05, Dunnett) than the non-treated (*p* < 0.0001, Dunnett), but shifting to more negative values. Q loading produced a similar effect but with smaller changes in the hydrodynamic parameters. RES-loaded nanoparticles showed significant statistical differences in the hydrodynamic parameters before irradiation (*p* < 0.0001, Dunnett), but the differences disappeared after irradiation. The ζ values were closer to the limit value that ensures the hydrodynamic stability of the suspension due to the high electrostatic repulsion of the particles. All these values agree with those previously published [24,25,26].

The morphology of the nanoparticles was studied by FESEM, confirming that all SFN-PPh were globular particles and quite homogeneous in size (see Figure 5) but showed greater aggregation than SFN.

Thus, while the dried unloaded SFN showed a smoother surface and remained as individual nanoparticles after the sample preparation (drying and coating with a Pd layer), the morphology of the nanoparticles was similar to that reported in previous works [24,25,26] and appeared as aggregates, presenting a rough surface. The γ-irradiation applied as sterilization method did not produce significant changes neither in morphology nor aggregation of the SFN-PPh, as can be seen in Appendix A (Appendix A), confirming previous published results [21].

Although the ATR-FTIR spectra of the loaded samples showed a similar full-range peaks profile as the unloaded SFN (Figure 6a), when we focused our attention on the amide regions of the spectra of the loaded particles, as the most informative of the secondary structure of the fibroin or the presence of PPh, some differences were found. In this region, the contributions of the Q signals to the SFN spectrum were more marked, increasing with the PPh loading content (Figure 6a). 

Taking into account the ATR-FTIR, we could assign the β-sheet conformation to SF in the loaded nanoparticles. Indeed, although the SFN-Q spectra showed a broader peak at ~1620 cm^−1^ compared to the unloaded nanoparticles (1624 cm^−1^), this contribution was due to the very closed carbonyl peak of the loaded PPh (1608 cm^−1^). Other significant peaks of the Q raised progressively from the SFN spectrum (Figure 6b). The resolved peaks of either Q or SFN can be found in the respective Appendix A (Appendix A).

### 3.3. Cell Viability Assay

In order to determine the safety and potential use of irradiated SFN-PPh, cell viability was examined by MTT assays. Nanoparticle suspensions at 100 μg·mL^−1^ and 500 μg·mL^−1^ (equivalent to 16 and 80 μg·mL^−1^ of each loaded PPh) were tested (Figure 7a) and compared to the effect of the equivalent concentrations of free PPh under the same conditions (Figure 7b). Our results showed that there were no significant differences in cell viability between cells exposed to SFN-PPh and those exposed to the correspondent free compounds (Detailed values are presented in Appendix A). More specifically, the viability of LPS-treated macrophage HL-60 cells was not affected by the exposure to either SFN-PPh or free compounds at the lowest concentration (16 μg·mL^−1^), finding cells viability values above 80%, which indicates no cytotoxicity. Similarly, cells exposed to 80 μg·mL^−1^ of free CUR, free Q and their SFN counterparts did not show significant differences in cell viability after 24 h compared to any of the control conditions (untreated, unloaded SFN). On the other hand, both SFN-RES and free RES greatly affected cell viability at its highest concentration (80 μg·mL^−1^; 350.5 μM), reducing it by 67% (*p* < 0.0001) and 77%, respectively (*p* < 0.0001). These results partially agree with previous reports where both RES and encapsulated RES showed cytotoxicity in a dose-dependent manner in other cell lines and confirm the cytotoxic effect of a high supra-pharmacological dose of both SFN-loaded and free RES (350.5 μM) also on our cell model.

As expected, irradiated unloaded SFN at 100 μg·mL^−1^ and 500 μg·mL^−1^ displayed good biocompatibility, showing no cytotoxicity on the macrophage HL-60 cells [21]. Furthermore, there was a significant increase in the metabolism of cells exposed to the highest concentration of unloaded SFN (*p* = 0.049; Figure 7a). In addition, similar results were obtained for cell viability in those cells exposed to non-irradiated SFN-PPh (Appendix A). Overall, these results show that sterilization through irradiation is a safe and efficient method for PPh encapsulation, which could facilitate their clinical use in the future.

### 3.4. In Vitro Effect of Polyphenol-Loaded SFN on Pro-Inflammatory Cytokines Production by Human Macrophage HL-60 Cells

PPh such as CUR, Q and RES are known to exhibit anti-inflammatory effects in several cell types [41,42,43]. Thus, the suppression of pro-inflammatory markers IL-6 and TNF-α release was evaluated in LPS-induced HL-60 macrophages exposed to free and SFN-PPh at a pharmacological dose of 16 μg·mL^−1^, for which no cytotoxic effect was previously observed. Compared to untreated control cells, cells treated with LPS showed a significant increase in IL-6 production (Figure 8a,b). As expected, all three compounds significantly reduced the IL-6 production induced by LPS at a dose of 16 μg·mL^−1^ (*p* < 0.05) in their free form, and this lowering effect was also observed when cells were treated with their corresponding SFN-encapsulated forms.

Furthermore, both free Q and SFN-Q significantly reduced IL-6 release compared to the untreated control. Interestingly, in contrast to free CUR (Figure 8b), SFN-CUR was also able to significantly further reduce the production of IL-6 compared to the untreated cells (*p* = 0.001, Figure 8a), reaching a level of inhibition like that observed with SFN-Q, although no differences were observed between free and SFN-Q treatment (Appendix A). This suggests that the use of SFN might result in an enhanced cellular uptake and bioavailability of the Q. On the other hand, none of the applied treatments showed a remarkable effect on TNF-α production. Furthermore, although the production of this cytokine following LPS treatment showed a rising trend, it did not reach statistical significance (Figure 8c,d). Nonetheless, all treatments (free and loaded compounds) showed a trend of getting TNF-α concentrations down to basal levels, and a further mild but not significant decrease in TNF-α was observed between LPS alone and LPS plus SFN-RES or free RES-treated cells (Figure 8c,d).

Finally, in order to confirm the uptake of SFN-PPh by macrophages, we leveraged the naturally occurring fluorescence of CUR by excitement at 488 nm. Thus, the distribution of SFN-CUR inside the cells was observed by fluorescent microscopy. As shown in Figure 9, cells exposed to SFN-CUR showed strong green fluorescence indicating a significant intracellular uptake of SFN-CUR into the macrophages.

These results support the idea that has been gaining interest in recent years regarding the utility of SFN as drug carriers due to their optimal physicochemical properties and biosafety. For instance, several studies have described that SFN enhance polyphenol bioavailability with a low cytotoxic effect associated, highlighting their potential use as therapeutics in different diseases such as cancer or inflammatory diseases [25,28,29,44,45]. However, the traditional preparation process performed until now has limited the potential use of SFN for large-scale assays, mainly due to its complexity and time consumption [46]. In this respect, we have described herein an optimized methodology that not only reduces the processing time, but also showed low cytotoxicity of SFN prepared with this new methodology in viability tests, as well as it was able to exert low immunogenicity.

This is in line with previous studies describing the immunogenicity of SFN obtained through different processes and in different formats [46,47,48]. Here we used an in vitro model of human macrophages-exposed SFN-PPh, and our results were consistent with those from other studies using both mouse cell lines and animal models describing their immunomodulatory effect through the reduction in inflammatory markers such as IL-6 and TNF-α [29,44,45,47,48]. Furthermore, the response of human macrophages to SFN was consistently optimal when particles were treated to be sterilized, supporting their good compatibility and biosafety, which may indicate the therapeutic potential of such loaded SFN.

## 4. Conclusions

The optimization of the synthesis of the nanoparticles and the further polyphenol loading process have allowed us to obtain low water solubility biomolecules-loaded nanoparticles with a considerable reduction in their processing time, of nearly 50%, when compared to traditional methods. In addition, the process is reproducible, achieves high DLC and the loaded nanoparticles exert high anti-inflammatory activity, as demonstrated by the reduction in the release of pro-inflammatory cytokines by macrophage differentiated HL-60 cells.

## Figures and Tables

**Figure 1 pharmaceutics-15-00263-f001:**
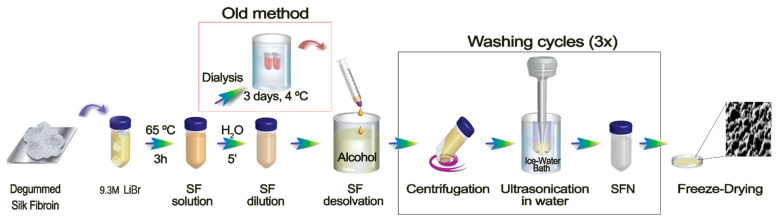
Scheme of the protocol developed for SF dissolution and consequent SFN preparation. Inside the red square is shown the traditional route and in the black square is highlighted the washing cycles necessary for the LiBr cleaning.

**Figure 2 pharmaceutics-15-00263-f002:**
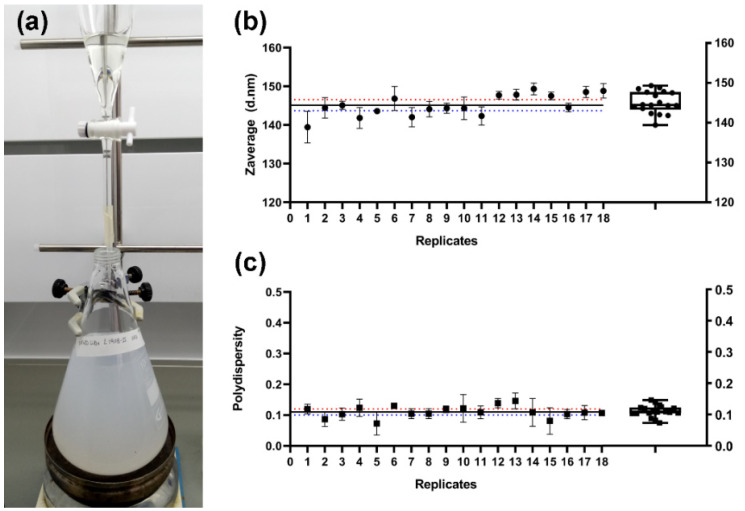
Preparation of the silk fibroin nanoparticles. (**a**) Photograph of the SF desolvation in methanol. (**b**,**c**) Values of the hydrodynamic characteristics of the prepared SFN (six replicates/batch) presented as mean ± SD of three independent measurements. (**b**) Zave (diameter in nm). (**c**) Polydispersity. Blue and red dotted lines represent the lower and upper 95% of the confidence interval (CI) of the mean, respectively. The black line represents the mean of all values (n = 18). Box and whiskers plot (10–90 percentile) is also included.

**Figure 3 pharmaceutics-15-00263-f003:**
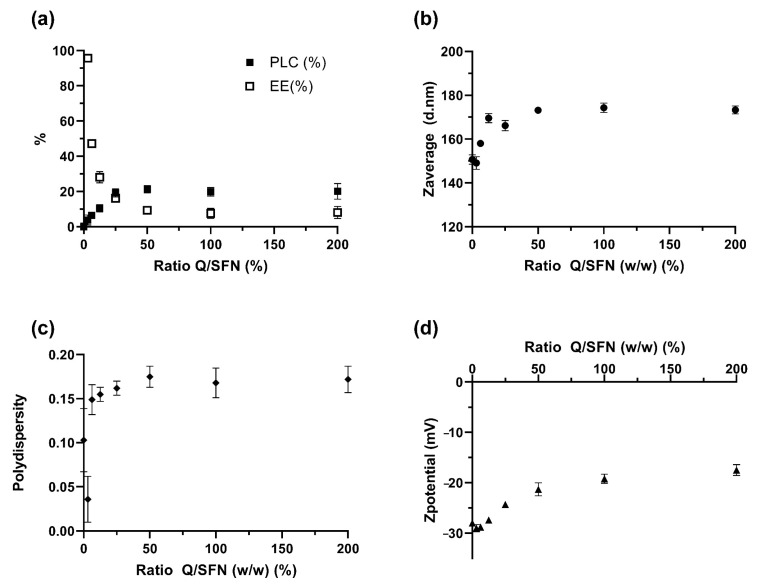
Effect of loading ratio Q/SFN in the PPh loading performance (**a**) and the hydrodynamic characteristics of the loaded nanoparticles (**b**–**d**).

**Figure 4 pharmaceutics-15-00263-f004:**
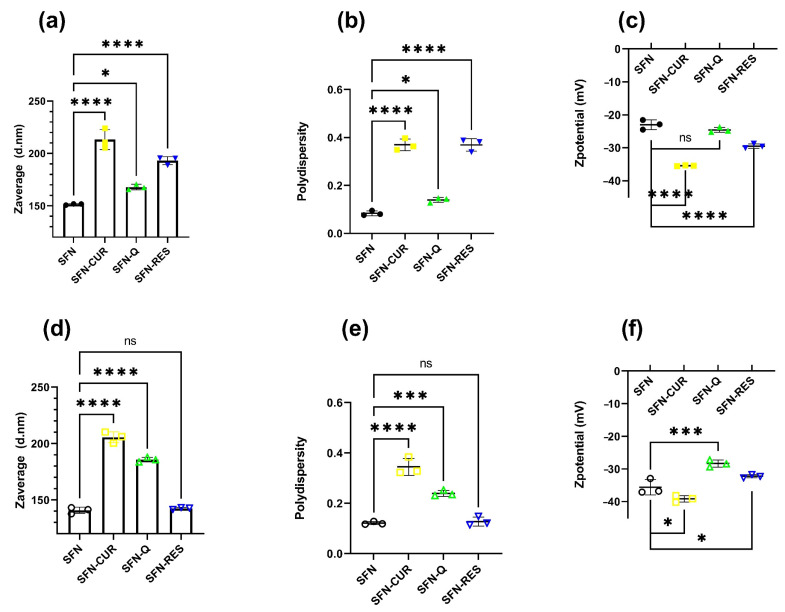
Hydrodynamic parameters of the aqueous suspensions of the SFN-PPh as prepared (**a**–**c**) and after the sterilization with γ-irradiation (2.5 kGy) (**d**–**f**). *p*-values were calculated using the two-way ANOVA test (considering *p* < 0.05 significant), followed by a Dunnet’s post hoc analysis, and are represented as * *p* < 0.05; *** *p* < 0.001; **** *p* < 0.0001.

**Figure 5 pharmaceutics-15-00263-f005:**
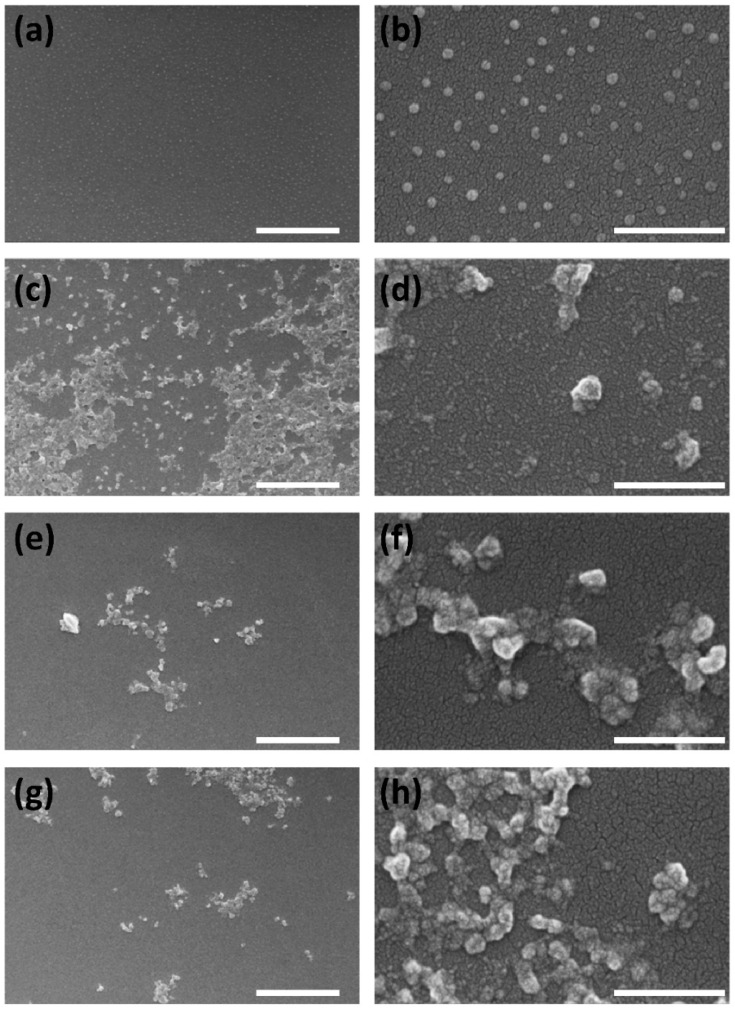
FESEM images of the nanoparticles used for the biological tests at different magnifications: SFN (**a**,**b**, as controls), SFN-CUR (**c**,**d**), SFN-Q (**e**,**f**) and SFN-RES (**g**,**h**). Scale bars 2 μm (**a**,**c**,**e**,**g**) or 400 nm (**b**,**d**,**f**,**h**).

**Figure 6 pharmaceutics-15-00263-f006:**
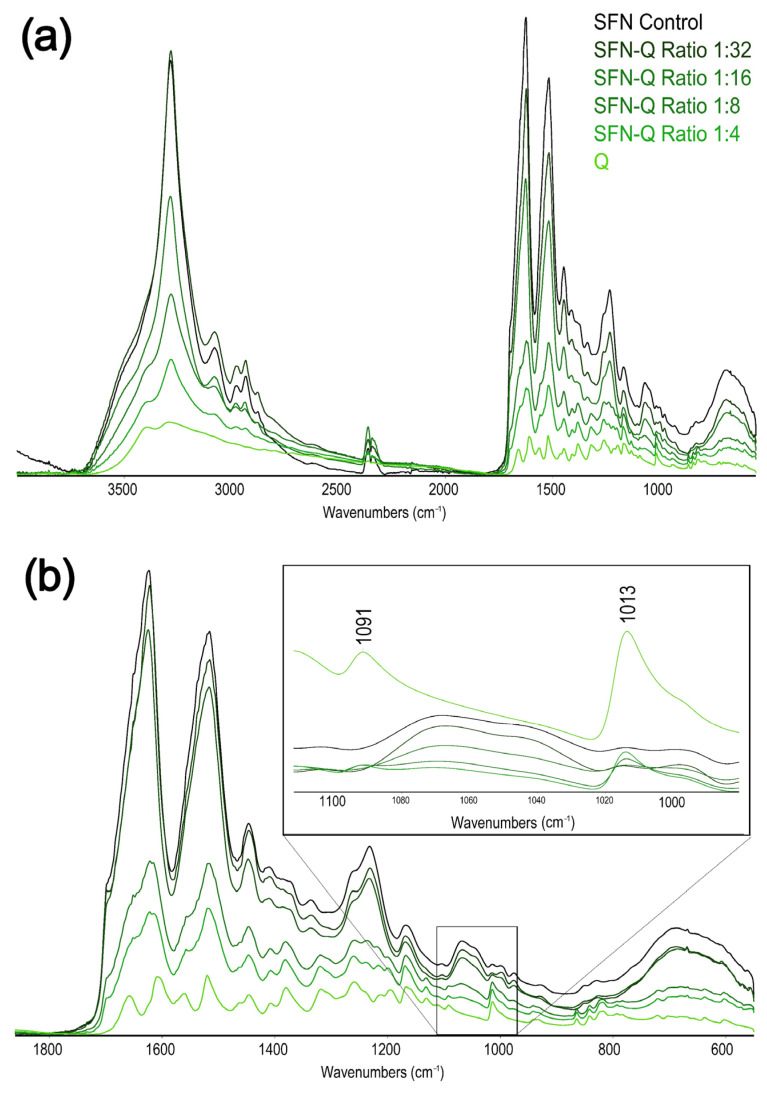
ATR-FTIR spectra of the prepared SFN-Q at different Q/SFN ratios. (**a**) Full spectra. (**b**) Region 1800–600 cm^−1^ of the spectra with an inset of the region 1750–100 cm^−1^ highlighting the quercetin signals.

**Figure 7 pharmaceutics-15-00263-f007:**
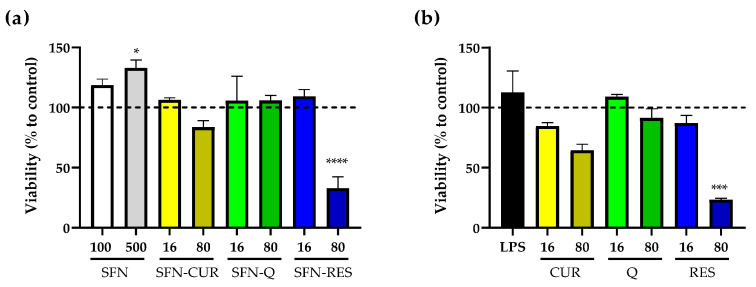
Effect of SFN-PPh and free PPh on the cell viability of human macrophage HL-60 differentiated cells measured by MTT assays (n = 3, three independent experiments performed in triplicate). (**a**) Cell viability in the presence of 100 or 500 µg·mL^−1^ of nanoparticles, equivalent to 16 or 80 µg·mL^−1^ of CUR, Q and RES according PLC. (**b**) Cell viability of HL-60 cells treated with free PPh (CUR, Q and RES) at the equivalent concentrations (16 and 80 µg·mL^−1^). Data are presented as % of cell viability ± SE of LPS-stimulated (0.1 µg·mL^−1^) HL-60 differentiated cells. Statistical differences were calculated using the two-way ANOVA test, followed by Dunnet’s post hoc analysis and represented as * *p* < 0.05; *** *p* < 0.001; **** *p* < 0.0001. All data presented are normalized compared to untreated cells (dash line, 100% cell viability). LPS, lipopolysaccharide; SFN, silk fibroin nanoparticles; PPh, polyphenol; SFN-CUR, SFN-curcumin loaded; SFN-Q, SFN-quercetin loaded; SFN-RES, SFN-resveratrol loaded; CUR, curcumin; Q, quercetin; RES, resveratrol.

**Figure 8 pharmaceutics-15-00263-f008:**
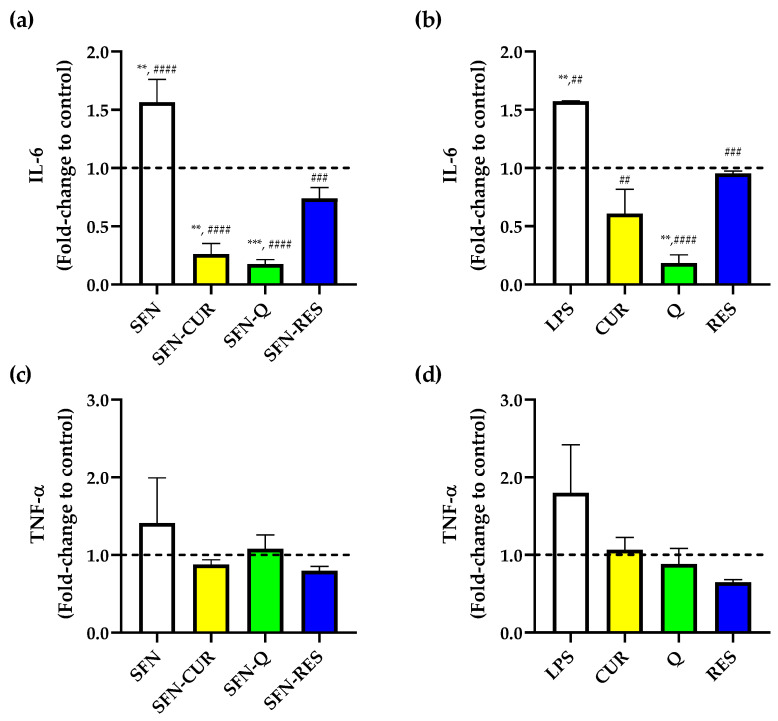
Effect of SFN-PPh and free compounds on cytokine release from the human macrophage HL-60 differentiated cells measured by ELISA. (**a**) IL-6 levels in SFN-PPh-treated cells, (**b**) IL-6 levels of free PPh-treated cells, (**c**) TNF-α levels of SFN-PPh-treated cells and (**d**) TNF-α levels in free PPh-treated cells (macrophage HL-60 differentiated cells under LPS-stimulated (0.1 µg·mL^−1^) conditions in the presence of nanoparticles (SFN or SFN-PPh (CUR, Q or RES) at 100 µg·mL^−1^) or free PPh (CUR, Q and RES) at the equivalent concentration (16 µg·mL^−1^) according the PLC. *p*-values were calculated using the two-way ANOVA test (considering *p* < 0.05 significant), followed by a Dunnet’s post hoc analysis. Values are expressed as fold-change to untreated cells (dash line). Results are represented as mean ± SEM from three different ELISA assays with three replicates performed in each assay condition. ** *p* < 0.01, *** *p* < 0.001 between treatments and untreated cells. ^##^
*p* < 0.01, ^###^
*p* < 0.001, ^####^
*p* < 0.0001 between treatments and LPS-stimulated control. LPS, lipopolysaccharide; IL-6, interleukin 6; TNF-α, tumor necrosis factor alpha; SFN, silk fibroin nanoparticles; PPh, polyphenols; SFN-CUR, SFN-curcumin loaded; SFN-Q, SFN-quercetin loaded; SFN-RES, SFN-resveratrol loaded; CUR, curcumin; Q, quercetin; RES, resveratrol.

**Figure 9 pharmaceutics-15-00263-f009:**
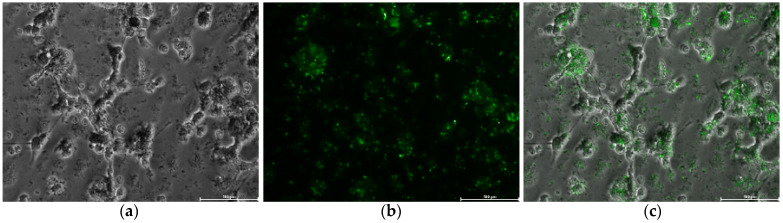
Human macrophage-like HL-60 cells after 24 h of the treatment with SFN-CUR at 0.1 mg·mL^−1^: (**a**) white field; (**b**) green channel; (**c**) merge. Scale bar = 100 μm.

## Data Availability

The data presented in this study are available on request from the corresponding author.

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
