# Peer review of "Optimizing the Preparation of Silk Fibroin Nanoparticles and Their Loading with Polyphenols: Towards a More Efficient Anti-Inflammatory Effect on Macrophages"

_pharmaceutics, 2023, doi:10.3390/pharmaceutics15010263_

Round 1

Reviewer 1 Report

The paper ‘Optimizing the preparation of silk fibroin nanoparticles and their loading with polyphenols: towards a more efficient anti- 3 inflammatory effect on macrophages’ can be accepted after major revision. The following comments should be addressed.

Comments:

1. There are many papers which have worked with polyphenol and silk fibroin nanoparticle. Novelty can’t be mentioned.

2. What is LiBr? Sudden abbreviation is not fine.

3.  Why the phytoconstituent is loaded passively? Is it possible to load maximum amount of it by this method.?

4. Spectroscopic determination of polyphenol should be shown in the manuscript?

5. Instead of several works on polyphenol with SF; no references were cited such as https://doi.org/10.3390/antiox9010085.

6. Some works on Phytomedicine should be more in discussion such as https://doi.org/10.3390/polym13183169; https://doi.org/10.1016/j.molliq.2018.02.024;

7. During indirect phytoconstituent loading, how the solution was prepared?

Author Response

Please read the attached file "Answer to Reviewer1.pdf.

Reviewer 2 Report

The authors have presented the experimental results very well and understandably. The illustrations are clear and the conclusions are reasonable.

Author Response

Answer to Reviewer 2: The authors would like to thank the reviewer for his/her positive feedback on the manuscript and kind words.